METHODS

# mbtransfer: Microbiome intervention analysis using transfer functions and mirror statistics

**Kris Sankaran** [1] *, **Pratheepa Jeganathan** [2]

**1** Department of Statistics, University of Wisconsin - Madison, Madison, Wisconsin, United States of America, **2** Department of Mathematics & Statistics, McMaster University, Hamilton, Ontario, Canada

\* ksankaran@wisc.edu

## Abstract

Time series studies of microbiome interventions provide valuable data about microbial ecosystem structure. Unfortunately, existing models of microbial community dynamics have limited temporal memory and expressivity, relying on Markov or linearity assumptions. To address this, we introduce a new class of models based on transfer functions. These models learn impulse responses, capturing the potentially delayed effects of environmental changes on the microbial community. This allows us to simulate trajectories under hypothetical interventions and select significantly perturbed taxa with False Discovery Rate guarantees. Through simulations, we show that our approach effectively reduces forecasting errors compared to strong baselines and accurately pinpoints taxa of interest. Our case studies highlight the interpretability of the resulting differential response trajectories. An R package, mbtransfer, and notebooks to replicate the simulation and case studies are provided.

**Data Availability Statement:** Software implementing the methodology is available on GitHub at https://go.wisc.edu/crj6k6. Code and data to reproduce simulation experiments are available on GitHub at https://go.wisc.edu/dxuibh.

## Author summary

Effectively controlling dynamic microbiomes has remained a major research challenge, primarily due to the interdependence between microbes and their sensitivity to environmental change. Tackling this challenge would advance microbiome engineering, with significant implications for healthcare, agriculture, and conservation. We introduce a flexible and statistically-principled approach to modeling microbe-microbe and microbe-environment relationships. We illustrate the methodology on case studies using microbiome time series datasets related to precision nutrition and women's health. We have released a software package, mbtransfer, to allow easy implementation of the methodology in other contexts where it is important to quantify intervention effects in temporally sampled data.

## Introduction

Microbiomes respond dynamically to environmental shifts. For example, changes in diet can rapidly alter diversity in the gut microbiome [1], childbirth can remodel the community state type of the vaginal microbiome [2], and the introduction of a pathogen can cause sub-communities to collapse [3]. Understanding these dynamics, together with how they fit into a

Preprocessed data and complete preprocessing scripts are available on figshare at https://figshare.com/projects/Microbiome_Intervention_Analysis/165559.

**Funding:** KS received support from grant number R01GM152744 from the National Institute of General Medical Sciences - https://www.nigms.nih.gov/. PJ received support from grant number 20016699 from the Faculty of Science at McMaster University. https://science.mcmaster.ca/. The funders played no role in the study design, data collection and analysis, decision to publish, or preparation of the manuscript.

**Competing interests:** The authors have declared that no competing interests exist.

microbiome's stable state, could guide the design of precise interventions to achieve desirable health or ecological outcomes [4, 5]. To this end, several models of microbial community dynamics have been developed to quantify environmental effects. A key model in this domain is the generalized Lotka-Volterra (gLV) model, which discretizes an ordinary differential equation for competitive predator-prey dynamics [5, 6]. Various extensions have been proposed to better account for microbiome-specific properties, like high-dimensionality and sparsity [7–10].

Specifically, denote the microbial community profile and environmental state at time $t$ by $\mathbf{y}(t)$ and $\mathbf{w}(t)$, respectively. The gLV supposes $\frac{\partial \mathbf{y}(t)}{\partial t} = A\mathbf{y}(t) + D\mathbf{w}(t) + \epsilon(t)$, and it can be fitted by log-transforming the observed taxonomic abundances $\log(1 + \mathbf{y}_t)$ and fitting an elastic net regression of $\log(1 + \mathbf{y}_{t+1}) - \log(1 + \mathbf{y}_t)$ against $\mathbf{w}_t$. The main limitations of this model are (1) that it assumes linearity in the relationship of $\log(1 + \mathbf{y}_{t+1}) - \log(1 + \mathbf{y}_t)$ onto $\mathbf{w}_t$ and (2) it relies on a Markov assumption, referring only to the immediate past. Moreover, it does not support formal statistical inference. In contrast, MALLARD [9] and fido [10] are based on a multinomial logistic-normal autoregressive and multinomial logistic-normal Gaussian Process models. In fido, the Gaussian Process structure encourages sharing across timepoints, and the model supports probabilistic inference. MDSINE and MDSINE2 [7, 8] are negative binomial dynamical systems models that extend the gLV and focus on the discovery of interspecies interactions and perturbation effects. Autoregressive dynamics are clustered using a Dirichlet Process, and a Gaussian Process prior regularizes species abundance trajectories.

We propose a **m**icro**b**iome **transfer** function (mbtransfer) modeling workflow to provide a simple but expressive language for intervention analysis in dynamic microbiome communities. The key ingredients of our approach are transfer function models [11], which summarize community dynamics, and mirror statistics [12], which enable precise inference. Transfer functions relate an "input" series to an "output" one. These models were originally developed to support intervention analysis in time series data, for example, the impact of new emissions regulation on local ozone levels [11], and we adapt this framework to the high-dimensional microbiome setting. Mirror statistics are an approach to selective inference [13] that leverages data splitting to rank differential effects while controlling the False Discovery Rate (FDR), and we develop an instance of this algorithm using partial dependence profiles of the fitted transfer function models. This approach is analogous to recent proposals based on knockoffs [14, 15], but it does not depend on the simulation of knockoff features, which can be sensitive to misspecification. Taken together, transfer function models and mirror-based inference address central questions in microbiome intervention analysis:

1. Which taxa are the most affected by the intervention? Mirror statistics identify taxa with differential trajectories across counterfactual environmental conditions.

2. When are these taxa affected? We can distinguish between transient and sustained shifts in the community by simulating counterfactual trajectories from fitted transfer function models.

3. Which factors mediate the shift? Flexible transfer function models can detect interactions between interventions and environmental features.

We release an accompanying R package to support the estimation of transfer function models, testing for significant taxon-level effects, and simulation of counterfactual alternatives. The package's source code and documentation, including notebooks to reproduce the case studies, can be found at https://go.wisc.edu/crj6k6.

## Materials and methods

### Transfer function models

Transfer function models were introduced as a linear autoregressive model applied to two concurrent time series, a series $w_t \in \mathbb{R}$ that serves as the intervention and a series $y_t \in \mathbb{R}$ that changes in response [11]. We consider the generalization,

$$\mathbf{y}_t^{(i)} = \mathbf{f}(\mathbf{Y}_{t-1}^{(i)}, \mathbf{W}_t^{(i)}, \mathbf{z}^{(i)}) + \epsilon_t^{(i)} \tag{1}$$

where we have used the following notation (see also Fig 1A):

- $\mathbf{y}_t^{(i)} \in \mathbb{R}^J$: The microbiome community profile for subject $i$ at time $t$. Includes measurements across all $J$ taxa. Alternative normalizations can be applied before defining $\mathbf{y}_t^{(i)}$ (size factor, total sum scaling, centered log-ratio), and their influence is considered in the simulation study below.

- $\mathbf{Y}_{t-1}^{(i)} \in \mathbb{R}^{J \times P}$: The combined measurements from time $t-P$ to $t-1$ for subject $i$, which can be interpreted as a length-$P$ memory of past community profiles. Note that $\mathbf{Y}_{t-1}^{(i)} = [\mathbf{y}_{t-P}^{(i)} | \cdots | \mathbf{y}_{t-1}^{(i)}]$.

- $\mathbf{W}_t^{(i)} \in \mathbb{R}^{D \times Q}$: The strength of $D$ interventions from time $t - Q + 1$ to $t$ for subject $i$. Interpreted as the length-$Q$ memory of environmental interventions, including the current timepoint.

- $\mathbf{z}^{(i)} \in \mathbb{R}^S$: The characteristics of subject $i$ that do not vary over time.

- $\epsilon_t^{(i)} \in \mathbb{R}^J$: Random error in the taxonomic measurements for timepoint $t$ in subject $i$. For valid inference (Eq 3), this must be assumed symmetric about zero.

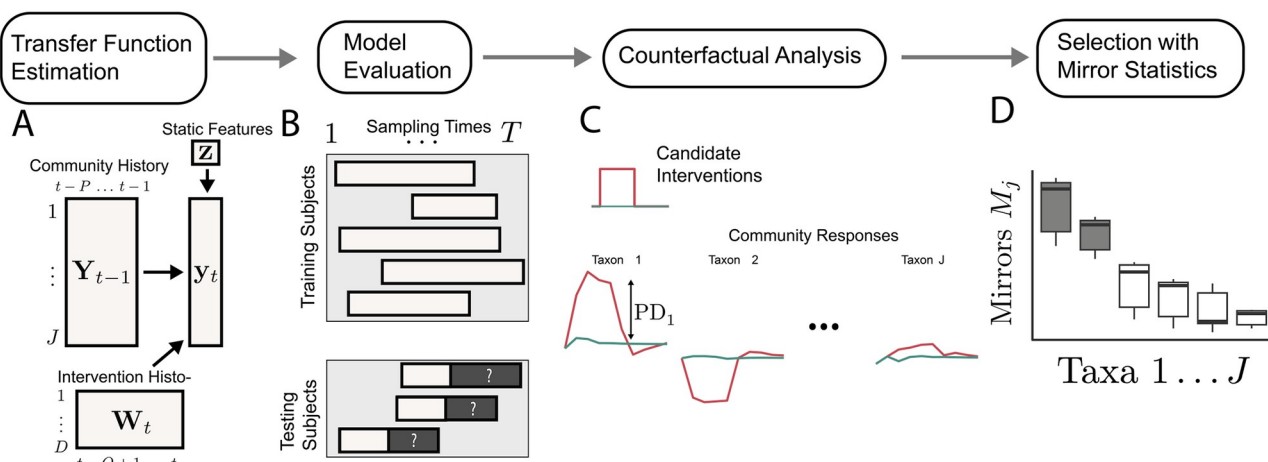

**Fig 1. Overview of the transfer function approach to modeling microbiome interventions.** (A) A transfer function model (Eq 1) is trained to forecast future community profiles. This model leverages past community data, past and current intervention information, and static subject-level characteristics. (B) Forecasts on held-out subjects are used to evaluate model performance, potentially guiding model improvements. (C) The trained models are used to simulate counterfactual trajectories, supporting the study of hypothetical interventions. Multiple interventions can be applied concurrently, and they may be real-valued. (D) To identify taxa sensitive to the interventions, partial dependence effects from simulated trajectories are used to calculate mirror statistics.

The model in Eq 1 can summarize and simulate intervention effects on microbial communities, accounting for baseline profiles and mediating host features.

What relationships are expressible within this model class? To answer this question, we separately consider the inputs and the model structure. We train each coordinate $f_j$ of $\mathbf{f}$ to predict $y_{tj}$ (taxon $j$'s abundance at time $t$) using flattened prior abundances $\mathbf{Y}_{t-1}$, perturbations $\mathbf{W}_t$, and static features $\mathbf{z}$. We supplement this input with interaction terms identified by the xyz algorithm [16] as potentially predictive of future abundance. These interaction terms allow inputs to modulate one another, analogous to interactions in ordinary linear models. We estimate each $f_j$ using linear boosting applied to these inputs. Note that the learned $\hat{f}_j$ depends on the number of boosting iterations and the learning rate, and we have omitted this dependence for ease of notation. Linear boosting can be understood as a regularized linear model learned through an additive basis expansion [17, 18]. While components are linear, the xyz-derived interactions between $\mathbf{Y}_{t-1}$, $\mathbf{W}_t$, and $\mathbf{z}$ induce nonlinearity with respect to the original inputs, reflecting the degree of the interaction. The fitted $\hat{f}_j$ can be analyzed to understand the influence of perturbations and other taxa on the abundance of taxon $j$. For example, in the predator-prey dynamics example below and in our definition of mirror statistics, we apply partial dependence profiles.

For estimation, we tile sample trajectories into disjoint time windows. For each window, we extract aligned taxonomic $\mathbf{Y}_{t-1}^{(i)}$ and intervention $\mathbf{W}_t^{(i)}$ samples. These are then vectorized and combined with $\mathbf{z}^{(i)}$ to form a $\mathbb{R}^{PJ+QD+S}$-dimensional feature vector. Once trained, the estimates $\hat{\mathbf{f}}$ can be used to simulate expected counterfactual trajectories under hypothetical interventions $\tilde{\mathbf{W}}_{t+1}$ given initial compositions $\tilde{\mathbf{Y}}_t$ and subject characteristics $\tilde{\mathbf{z}}$. The one-step forecast is:

$$\hat{\mathbf{y}}_{t+1} = \hat{\mathbf{f}}(\tilde{\mathbf{Y}}_t, \tilde{\mathbf{W}}_{t+1}, \tilde{\mathbf{z}})$$

and longer time horizons can be forecast by substituting intermediate predictions:

$$\hat{\mathbf{y}}_{t+h} = \hat{\mathbf{f}}(\hat{\mathbf{Y}}_{t+h-1}, \tilde{\mathbf{W}}_{t+h}, \tilde{\mathbf{z}}). \tag{2}$$

Note that we use observed input profiles whenever possible. That is, when $\hat{\mathbf{Y}}_{t+h-1}$ includes times $t' \leq t$, we use the original values $\mathbf{y}_{t'}$ and no predictions are necessary. This formulation can detect nonlinear relationships between past microbial community profiles, interventions, and host features with taxon $j$'s current abundance. Further, it can detect interaction effects between covariates that improve predictive power, and these interactions are often of independent interest.

## Mirror statistics

**Background.** The transfer function model in Eqs 1 and 2 summarizes the effects of interventions $\mathbf{W}_t$ on current and future community profiles $\mathbf{y}_t$ and $\mathbf{y}_{t+h}$. However, they do not provide inferential guarantees on the existence of either immediate or delayed effects following the intervention. Therefore, we propose a mirror statistics approach [12], which supports variable selection under minimal assumptions, symmetry under the null hypothesis, and weak dependence across tests, whose plausibility is discussed below.

Before discussing its application within our context, we briefly review multiple testing with mirror statistics. Suppose that data have been drawn from a linear model $\mathbf{y} = \mathbf{X}\beta + \epsilon$ and our goal is to identify the nonzero elements of $\beta \in \mathbf{R}^J$. The intuition behind mirrors is to split the data into $\mathcal{D}^{(1)} = (\mathbf{X}^{(1)}, \mathbf{y}^{(1)})$ and $\mathbf{D}^{(2)} = (\mathbf{X}^{(2)}, \mathbf{y}^{(2)})$ and check for agreement in the estimates $\hat{\beta}^{(1)}$ and $\hat{\beta}^{(2)}$ across splits. For coordinate $j$, consider the statistic $M_j = \text{sign}\left(\hat{\beta}_j^{(1)}\hat{\beta}_j^{(2)}\right)\left[\left|\hat{\beta}_j^{(1)}\right| + \left|\hat{\beta}_j^{(2)}\right|\right]$. Observe

that if there is a true effect, then we expect $M_j$ to be positive because the signs across splits should agree. In contrast, under the null, we will assume $M_j$ is symmetric about 0—our estimator $\hat{\beta}_j$ should not have a preference for either negative or positive estimates. Observe that $\left[\left|\hat{\beta}_j^{(1)}\right| + \left|\hat{\beta}_j^{(2)}\right|\right]$ gives the magnitude of feature $j$'s effect, and it is natural to declare significance using the rule: reject $j$ if $M_j > t$ for some threshold $t$.

Mirror statistics uses this symmetry assumption to estimate the false discovery rate for any given $t$. Specifically, if $M_j$ is symmetric about 0 under the null, then the number of null $M_j$ above $t$ should roughly match the number of null $M_j$ below $-t$. Since we expect few true effects to have $M_j < -t$, we can estimate the false discovery rate of the above rule using $\widehat{\text{FDR}}(t) = \frac{|\{j:M_j>t\}|}{|\{M_j<-t\}|}$. The mirror algorithm selects the smallest $t$ such that $\widehat{\text{FDR}}(t)$ lies below the target level. Finally, it is possible to repeat this procedure across many random splits and aggregate evidence across them. See Fig B in S2 Text for a graphical summary. Viewed more abstractly, the first assumption of this algorithm is that $M_j$ is symmetric about 0 under the null. While intuitive for linear models, this could hold for other problems, and the specific form of the statistic is relevant only for validity, not power. Less obviously, the estimate $\widehat{\text{FDR}}(t)$ relies on the assumption that the $M_j$ are at most weakly dependent. In the experiments below, we empirically investigate mirror statistics' robustness to violations of this assumption.

At a high level, this algorithm benefits from transforming testing to regression. The assumptions above are still weaker than the parametric assumptions that accompany many multiple testing procedures and potentially offer an avenue around the elevated false discoveries that can arise when distributional assumptions fail [19]. Moreover, mirror statistics can serve as a meta-algorithm, wrapping more specialized estimation routines as long as appropriate mirror statistics can be derived. We illustrate this for transfer functions below.

**Application to transfer functions.** As in the discussion above, we first randomly split subjects into subsets, $\mathcal{D}^{(1)}$ and $\mathcal{D}^{(2)}$. For each split $s$, we estimate models $\hat{\mathbf{f}}^{(s)}$ using $d_t^{(i)} = (\mathbf{y}_{t+1}^{(i)}, \mathbf{Y}_t^{(i)}, \mathbf{W}_{t+1}^{(i)}, \mathbf{z}^{(i)})$ for subjects $i$ in that split and nonoverlapping segments beginning at times $t$. Next, we estimate the counterfactual difference between interventions $\tilde{\mathbf{W}}_t = \mathbf{1}_Q$ and $\tilde{\mathbf{W}}_t = \mathbf{0}_Q$ for each taxon:

$$\text{PD}_j^{(s)} = \frac{1}{|\mathcal{D}^{(s)}|} \sum_{d_t^{(i)} \in \mathcal{D}^{(s)}} \left[ \hat{f}_j^{(s)}\left(\mathbf{Y}_t^{(i)}, \mathbf{1}_Q, \mathbf{z}^{(i)}\right) - \hat{f}_j^{(s)}\left(\mathbf{Y}_t^{(i)}, \mathbf{0}_Q, \mathbf{z}^{(i)}\right) \right].$$

This equation is a partial dependence profile of $\hat{f}_j^{(s)}$ [17, 20] measuring the effect of $Q$ consecutive interventions on taxon $j$. For isolated interventions, we can use $(0, \ldots, 0, 1, 0, \ldots, 0)$ instead of $\mathbf{1}_Q$. Intuitively, this approximates the difference in short-term forecasts $\hat{f}_j^{(s)}$ of taxon $j$'s trajectory under contrasting interventions, with the empirical distributions of $\mathbf{Y}_t^{(i)}$ and $\mathbf{z}^{(i)}$ used as plug-in estimates for the unknown population distribution. Finally, we define the mirror statistic:

$$M_j = \text{sign}\left(\text{PD}_j^{(1)}\text{PD}_j^{(2)}\right)\left[\left|\text{PD}_j^{(1)}\right| + \left|\text{PD}_j^{(2)}\right|\right],$$

which gauges the consistency between estimated effects across separate splits. When there are no true intervention effects on taxon $j$ and $\epsilon_t^{(i)}$ is symmetric around 0, then $\text{PD}_j^{(s)}$ is also symmetrically distributed around 0, thus satisfying the mirror statistics assumption. Specifically, in the absence of an intervention effect, any differences between $\hat{f}_j^{(s)}(\mathbf{Y}_t^{(i)}, \mathbf{1}_Q, \mathbf{z}^{(i)})$ and

$\hat{f}_j^{(s)}(\mathbf{Y}_t^{(i)}, \mathbf{0}_Q, \mathbf{z}^{(i)})$ are due to noise. In contrast to the symmetry assumption, weak dependence is less likely to hold in practice because many taxa may be expected to respond similarly to interventions. Since the theory of mirror statistics requires weak dependence to guarantee FDR control, and since violation seems likely, our simulations include a setting with phylogenetically correlated hypotheses.

Given mirror statistics $M_j$ and a threshold $t$, we estimate the false discovery proportion using the same estimator discussed above:

$$\widehat{\mathrm{FDR}}(t) \quad = \frac{|\{j : M_j < -t\}|}{|\{j : M_j > t\}|}. \tag{3}$$

The choice of $t$ defines a selection set $\hat{J}_1$ of sensitive taxa, and to control the FDR at level $q$, we choose the largest $t$ such that $\widehat{\mathrm{FDR}}(t) \leq q$. We improve power by aggregating across multiple splits, following Algorithm 2 of [12]. Our examples use 25 splits, which is fewer than the 50 in [12], but which nonetheless provides sufficient power in our applications. For delayed effects, we define analogous $\mathrm{PD}_j^{(s),+h}$ and $M_j^{+h}$ using $h$-step ahead predictions (Eq 2), modifying the estimate in Eq 3 to use mirrors $M_t^{+h'}$ across all lags $h' \leq h$.

This methodology is implemented in the R package mbtransfer (https://go.wisc.edu/crj6k6), which includes functions for estimating transfer function models, simulating counterfactual trajectories, and performing inference with mirror statistics. Further, the package includes data structures and utilities for manipulating intervention data and visualizing estimated transfer functions and mirror statistics.

## Results

### Toy example

Can mbtransfer capture predator-prey dynamics? Using the seqtime package [21], we generated data from a Lotka-Volterra model [22]:

$$\dot{y}_1 \quad = b_1 y_1 - a_{12} y_1 y_2 \ (\text{Prey})$$
$$\dot{y}_2 \quad = -b_2 y_2 + a_{21} y_1 y_2 \ (\text{Predator})$$

with parameters $b_1 = 2$ and $b_2 = a_{12} = a_{21} = 1$ for 10 time units at discretization $\Delta = 0.2$. This system exhibits oscillatory dynamics. Further, we perturb the prey's growth rate during the intervals $t \in [3, 4]$ and $t \in [8, 9]$, changing $b_1 = 2$ to 1. This intervention on the prey's growth rate indirectly decreases the predator population size. mbtransfer's functional form allows it to capture both species interactions and perturbation effects. We can write the discretized, perturbed system as:

$$y_{1(t+\Delta)} = (1 + \mathbf{1}\{t \in \text{perturbed}\} + \Delta b_1 - \Delta a_{12} y_{2t}) y_{1t} \tag{4}$$

$$y_{2(t+\Delta)} = (1 - \Delta b_2 + \Delta a_{21} y_{1t}) y_{2t} \tag{5}$$

This has the form $\mathbf{y}_{t+\Delta} = \mathbf{f}(\mathbf{y}_t, \mathbf{w}_t)$ for an $\mathbf{f}$ with interaction effects between the two coordinates of $\mathbf{y}_t$ and between $y_{1t}$ and $\mathbf{w}_t = \mathbf{1}\{t \in \text{perturbed}\}$. These interactions are the reason we observe complex, oscillatory behavior in a simple two-species system.

We trained an mbtransfer model ($P = Q = 1$) on 500 randomly initialized ($\mathbf{y}_0 \sim \text{Unif}[0, 4]$) examples, reserving $t \in [0, 7]$ and $(7, 10]$ for training and forecasting, respectively. Fig 2 visualizes 8 subjects chosen to represent a range of performance regimes. Adjacent pairs of subjects represent the minimum, 25%, 75%, and maximum forecasting error, read from the top left to

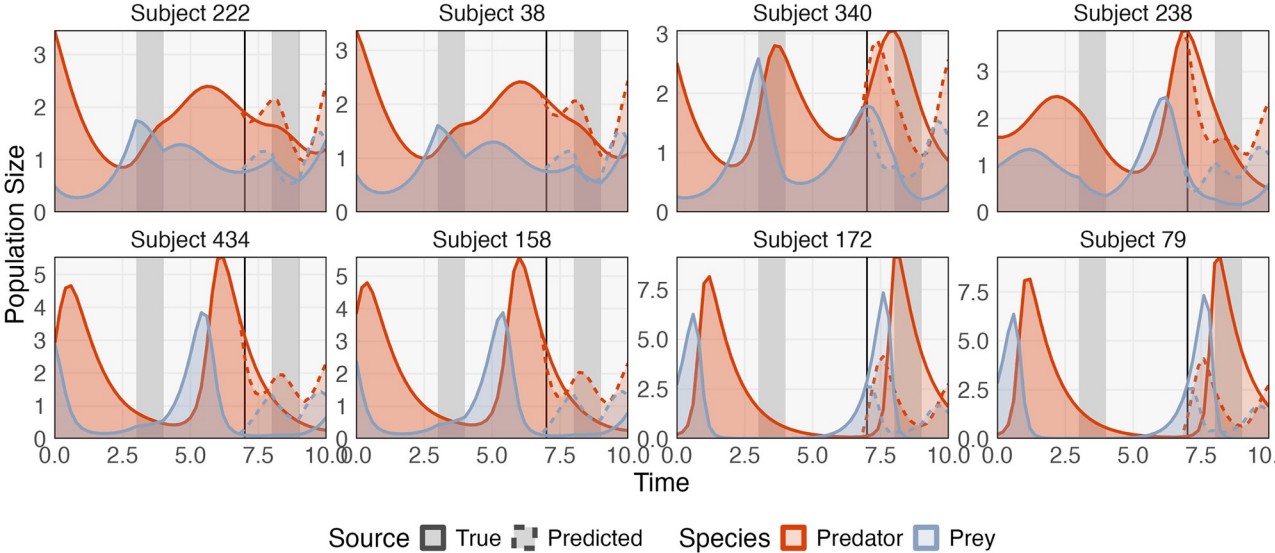

**Fig 2. Transfer functions applied to oscillatory predator-prey dynamics.** Panels give simulated predator and prey populations over 10 time units. The grey rectangles indicate periods during which the prey growth rate is decreased. A transfer function model is trained using data up to time 7, and forecasts are shown as dashed curves. Subjects have been chosen to represent low (top left) to high (bottom right) forecasting error rates. The forecasts accurately reflect the true perturbation effect and predator-prey relationship; however, they can compress time and dampen large peaks.

bottom right. Across regimes, mbtransfer learns the perturbation effect and the oscillatory predator-prey relationship. However, it generally "compresses" time, underestimating the period between peaks. Further, the model underestimates the size of the peaks in subjects with large populations. To better understand $\hat{\mathbf{f}}$, Fig C in S2 Text gives partial dependence profiles [17, 23]. These profiles represent the influence of each input, holding all others fixed, allowing them to disentangle species-species and perturbation-species relationships. Panels (A) and (E) reveal that the model understands that the perturbation immediately impacts the prey but not the predator population. Panels (B—C) and (E—F) relate current with predicted populations. For example, (C) shows that higher predator populations decrease the forecast prey population. A weak interaction effect is present in (B), with the slope decreasing when predator populations are larger. This is consistent with the first line of Eq 4. A related effect appears in (C), where slopes become more negative when the prey population is large. Ideally, these lines should all be exactly linear with gradually varying slopes. In contrast, our estimates are slightly nonlinear, and the interaction is only clear in large predator populations. This reflects a trade-off: without making stronger assumptions on the form of $\mathbf{f}$, larger datasets are needed to accurately estimate more subtle interactions.

## Simulation study

**Data generating mechanism.** In general, simulating realistic dynamic microbiome data is an open problem, especially when aiming to benchmark the relationship between microbes and their environment [24–27]. To ensure fairness in our evaluation, we simulated data from a transparent model that captures many microbiome properties while avoiding assumptions specific to any single model discussed below. Specifically, we adapted an autoregressive negative binomial factor model to reflect two challenges of microbiome data analysis:

- Phylogenetic correlation: Microbiome data often have correlated taxa due to shared evolutionary ancestry or occupation of similar ecological niches.

- Uneven sequencing depth: In a typical experiment, not all samples are sequenced to the same read depth. This necessitates sample-wise normalization.

The negative binomial distribution ensures sparsity and overdispersion, characteristics known to exist in microbiome data. The simulation data-generating mechanism is:

$$\mathbf{y}_t^{(i)}|\theta_t^{(i)}, \varphi, b^{(i)} \quad \sim \mathrm{NB}\left(b^{(i)}\exp\left(\theta_t^{(i)}\right), \varphi\right)$$

$$\theta_t^{(i)} \quad = \sum_{p=1}^P A_p \theta_{t-p}^{(i)} + \sum_{q=1}^Q \left(B_q + C_q \odot z^{(i)}\right)\mathbf{w}_{t-q}^i + \epsilon_t^{(i)}$$

$$b^{(i)} \quad \sim \Gamma(10, \lambda)$$

$$\epsilon_t^{(i)} \quad \sim \mathcal{N}(0, \Sigma)$$

In this context, $\theta_t^{(i)} \in \mathbb{R}^J$ is a vector, and NB denotes a negative binomial distribution applied element-wise to each taxon $j$ under a mean-dispersion parameterization. $A_p \in \mathbb{R}^{J \times J}$ captures the lag-$p$ competitive/cooperative dynamics between taxon pairs, while $B_q \in \mathbb{R}^{J \times D}$ represents the lag-$q$ effect of the $D$ interventions. $C_q$ represents an interaction between host characteristics and interventions, reflecting that intervention effects may be mediated by specific host features. $A$, $B$, and $C$ are chosen to be low-rank, and the detailed parameter generation process is detailed in Section A.1 in S1 Text.

The scaling factor $b^{(i)}$ mimics variation in sequencing depth. The mean and variance of $b^{(i)}$ are $\frac{10}{\lambda}$ and $\frac{10}{\lambda^2}$, and we set $\lambda$ to be either 10 or 0.1 to create more or less concentrated sequencing depths, respectively. The covariance $\Sigma$ is designed to capture phylogenetic correlation, with $\Sigma_{jj'} = (1 + d_{jj'})^{-\alpha}$, where $d_{jj'}$ is the cophenetic distance between taxa $j$ and $j'$ on a balanced binary tree, a stand-in for real phylogenetic structure. We consider $\alpha \in \{0.1, 10\}$ corresponding to high and low intertaxa correlation. Heatmaps of $\Sigma$ when $J = 200$ are given in Fig D in S2 Text. Note that even when $\Sigma$ is nearly diagonal, the coordinates of $\mathbf{y}_t^{(i)}$ are still correlated, a consequence of the low-rank structure of $A$ and $B$. The motivation for studying alternative dependence structures is to understand their impact on the validity of the mirror statistics-based selection mechanism, whose theoretical guarantees rely on weak dependence as $J$ tends to infinity.

A visualization of the trajectories for null and nonnull taxa is given in Fig 3. We generated 108 datasets with varying numbers of taxa, the proportion of null taxa (those unaffected by interventions), and signal strengths, see Section A.2 in S1 Text.

**Model settings and metrics.** We gathered performance metrics associated with alternative normalization, forecasting, and inference approaches. For normalization, we considered working with the original, untransformed data, the DESeq2 size-factor normalized data [28], and the size-factor normalized data followed by an asinh transformation [29, 30]. The latter two transformations account for potentially different library sizes across simulated samples and the fact that microbiome data can be highly skewed. For forecasting, we applied MDSINE2, fido, and mbtransfer. The MDSINE2 and fido models are chosen in favor of their predecessors MDSINE and MALLARD; they are summarized in Section C in S1 Text. We caution that fido was designed for relative abundance data and may not be suitable for normalized absolute abundances. However, due to limited options for microbiome intervention analysis that account for temporal structure, we have included it in the comparison. Both MDSINE2 and fido are Bayesian models, and we kept all priors at their software defaults. In practice,

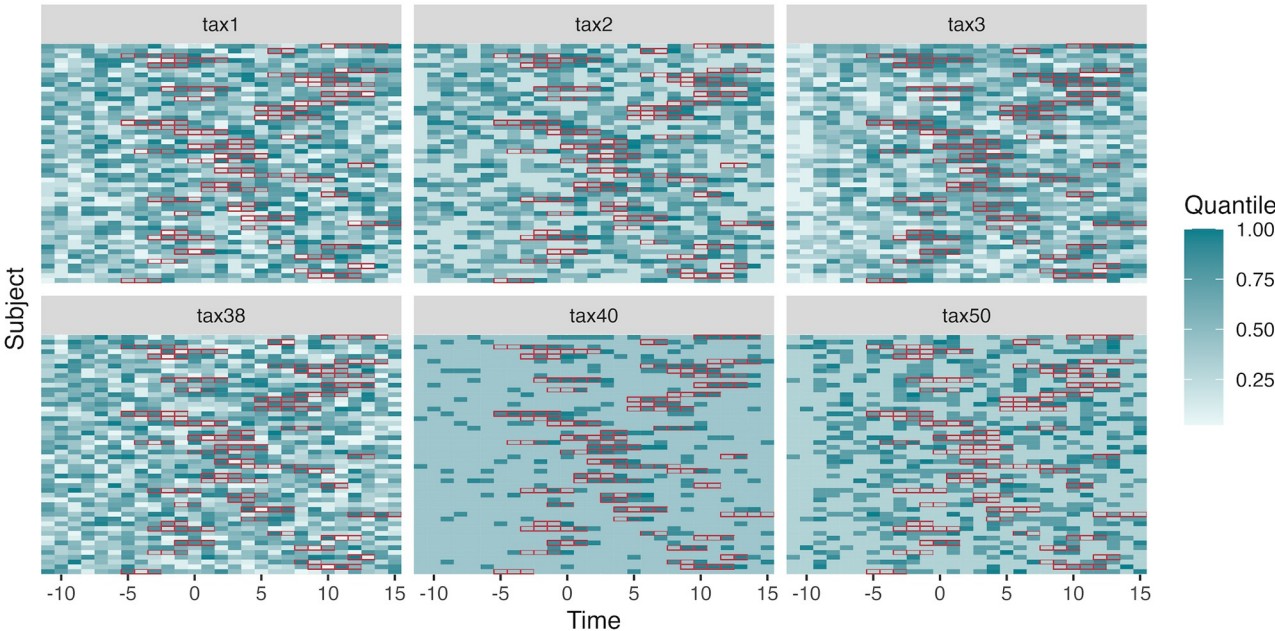

**Fig 3. Example data used in the simulation study.** Each panel displays one taxon's trajectories, with rows representing individual subjects. Tile colors encode abundances, which have been quantile transformed to support cross-taxa comparison. Red borders indicate the samples where the intervention is present. The first row of taxa (tax1—3) have nonnull effects, while the bottom row are all null. Note the potentially delayed intervention effects in nonnull taxa.

these priors could be adapted to the specific scientific problem at hand, and we acknowledge the inability of our simulation to evaluate this more mindful workflow.

For both fido and mbtransfer, we include two sets of hyperparameters to encourage longer vs. shorter temporal memories. For fido, we used an RBF kernel with hyperparameters set to either $\sigma = \rho = 0.5$ (short memory) or $\sigma = \rho = 1$ (long memory), and we applied mbtransfer with either $P = Q = 2$ (short memory) or $P = Q = 4$ (long memory). MDSINE2 forecasts are formed by integrating the learned dynamics over future timepoints, setting the initial conditions equal to the current test sample. Note that MDSINE2 was designed assuming the availability of qPCR data. Since qPCR data were unavailable for the case studies below, we have simulated qPCR values centered at 1e9, which is the same order of magnitude as examples available in the MDSINE2 documentation (https://go.wisc.edu/wuvfx4). We excluded MDSINE2 on runs with 400 taxa due to consistently long computation times ($> 72$ hours). For evaluation, we computed the mean absolute forecasting error across all taxa up to a time horizon of $h = 5$ on held-out subjects. Section A.3 in S1 Text provides details of the evaluation mechanism.

For inference, we compared the mirror algorithm to DESeq2 [28] with the formula $\sim$ w$_t$ * z, which tests for intervention effects, subject-level effects, and their interaction. DESeq2 is a negative binomial-based generalized linear model that has exhibited strong performance in differential abundance testing benchmarks [31]. This approach allows for intervention and subject-level effects but does not explicitly model microbiome dynamics. As an additional baseline, for each taxon, we applied two-sample $t$-tests to test for a change in mean between the four samples before and four samples after the start of the intervention. These relatively short windows were chosen because the effect often decays rapidly following the initial event. We adjusted $p$-values for multiple comparisons using the Benjamini-Hochberg procedure [32]. We evaluated inferential quality using false discovery proportions and power across

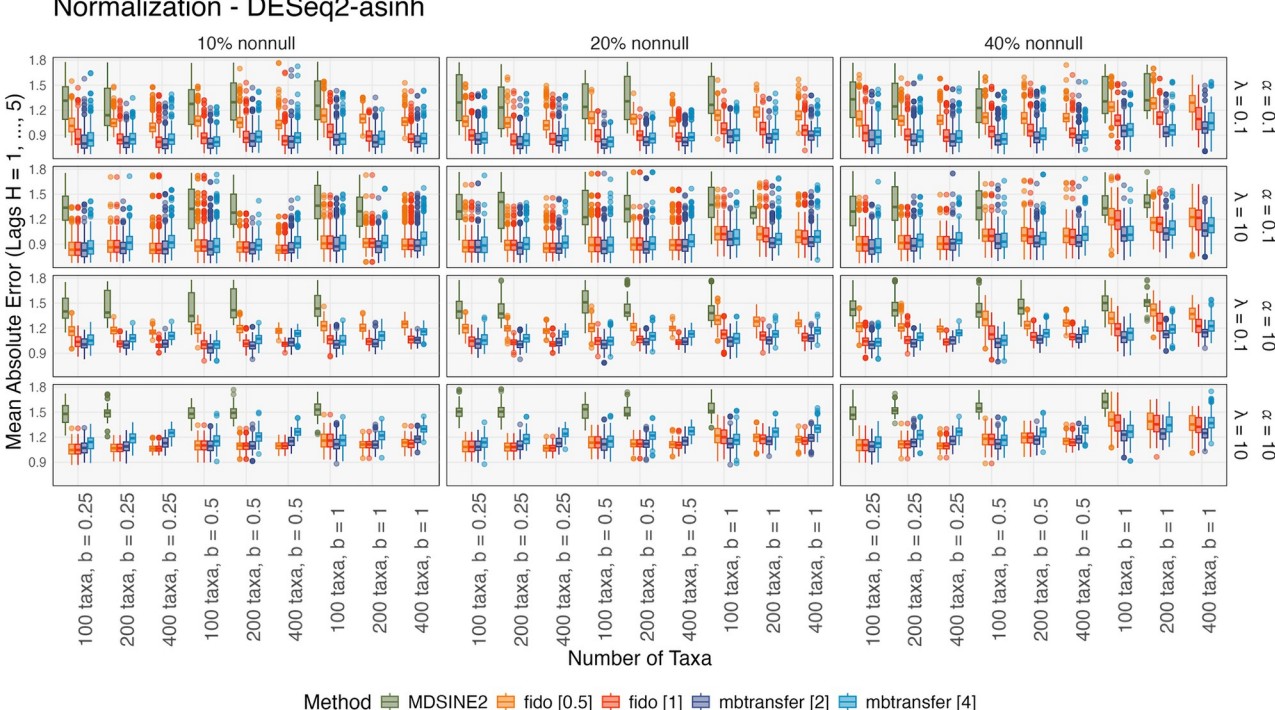

**Fig 4. Simulation forecasting errors for normalized data.** The *y*-axis shows the average $MAE_k$ across folds. Within panels, the signal strength and number of taxa increase from left to right. Column panels give the proportion of intervention-sensitive taxa and the signal strength. Rows distinguish between settings with different sequencing depth heterogeneity and phylogenetic correlation. Outliers (below 1.5×IQR or above 3×IQR of errors in fido and mbtransfer) are excluded. Runs that did not complete within 72 hours are omitted. The two hyperparameter settings of fido and mbtransfer perform similarly. The fido package is comparable to mbtransfer when the intervention strength is weak but deteriorates when the intervention is strong.

time lags. For delayed effects at lag $q$, we accounted both for taxa with nonzero entries of $B_q$ and taxa that were indirectly shifted by autoregressive links $A_p$ with taxa that are affected by the intervention at an earlier time. Formulas for the false discovery proportion and power across lags are given in Section A.3 in S1 Text.

**Evaluating forecasting performance.** Fig 4 summarizes cross-validated forecasting performance on DESeq2-asinh transformed data, showing that mean absolute error increases with the proportion of nonnull taxa and signal strength. This is due to higher variance shifts in $\mathbf{y}_t$ during interventions under these settings. MDSINE2 consistently performed worse than fido and mbtransfer. Fig 5 illustrates the prediction error for holdout subjects in one simulation setting, revealing that minor errors in MDSINE2's initial forecast become amplified at longer time horizons. Since MDSINE2 can only refer to one step in the past, it must have either exponential growth or decay until the community reaches its carrying capacity. While this behavior does not impact inferences for taxon-perturbation relationships, the main focus of [8], it restricts the usefulness of simulated hypothetical trajectories. In contrast, mbtransfer and fido can refer to longer historical windows, supporting more realistic intervention analyses: the second day of a microbiome intervention may have different consequences than the first.

Across methods, performance declines for larger $\alpha$. In this case, the phylogenetic correlation is weaker, and there is less signal to borrow from related taxa. The effect of sequencing depth on performance depends on specific method and data settings. Overall, mbtransfer's

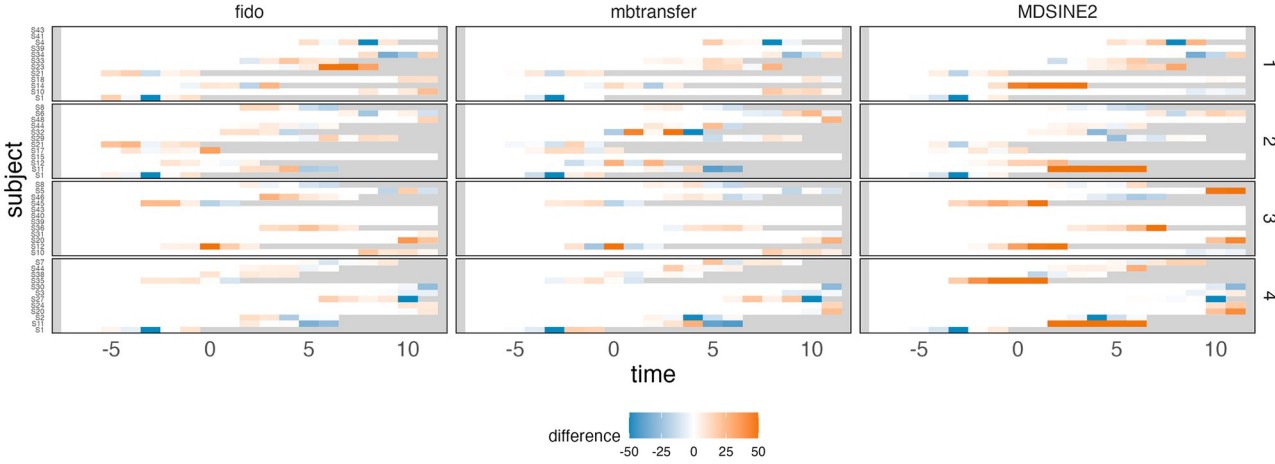

**Fig 5. Comparison of long-run forecasting residuals.** We average errors across all taxa and truncate those with a magnitude greater than 50. A comparison of forecasting residuals across four folds (rows) in one simulation run suggests that forward integrating the MDSINE2 model can lead to exponentially increasing forecasting errors.

performance relative to alternatives improves when sequencing depths are more variable, though forecasting becomes more difficult in this regime. When the intervention effect had a smaller magnitude or is limited to fewer taxa, fido and mbtransfer performed comparably. In other cases, mbtransfer was more accurate. We interpret this by noting that, despite its ability to incorporate interventions, fido's Gaussian Process assumption enforces temporal smoothness in the predictions. This prevented the model from capturing the sharp changes in abundance during strong interventions. However, as noted above, we have avoided extensive hyperparameter tuning or tailoring of flexible priors, and it is possible that such optimizations could improve performance.

Analogous results for alternative transformations are available in Figs E and F in S2 Text. When the data were not asinh transformed, the mbtransfer model performed worse than either MDSINE2 or fido. This reversal is consistent with the use of a squared-error loss in the underlying gradient boosting models, which is not adapted to count data. fido should be preferred if data must be modeled on the original scale. However, we note that transformations are often well-justified in microbiome analysis, and an increasing number of formal methods implement them [33–35]. Fig G in S2 Text gives the average computation time across folds. MDSINE2 was slower than either fido or mbtransfer. fido and mbtransfer had comparable computation times except when using DESeq2-asinh transformed data. In this setting, fido was faster, but mbtransfer was more accurate.

**Evaluating inferential performance.** For longer time horizons, all methods showed improved FDR control because more taxa become truly nonnull after instantaneous effects propagate across the community (Fig 6). DESeq2's performance is more influenced by the fraction of nonnull hypotheses, while mirrors are more influenced by the total number of taxa. For lag 1 effects, DESeq2 and the pre/post $t$-test fail to control the FDR at the specified level $q = 0.2$, likely a consequence of considering samples as independent when they are, in fact, temporally dependent.

For lag 2 effects, mirrors have slightly lower power than DESeq2. At this lag, both methods are conservative, except several DESeq2 runs with many taxa and few true nonnulls. Higher phylogenetic correlation ($\alpha = 0.1$) inflates the false discovery proportion for mirrors applied to unnormalized data. Nonetheless, for both high and low phylogenetic correlation, mirrors

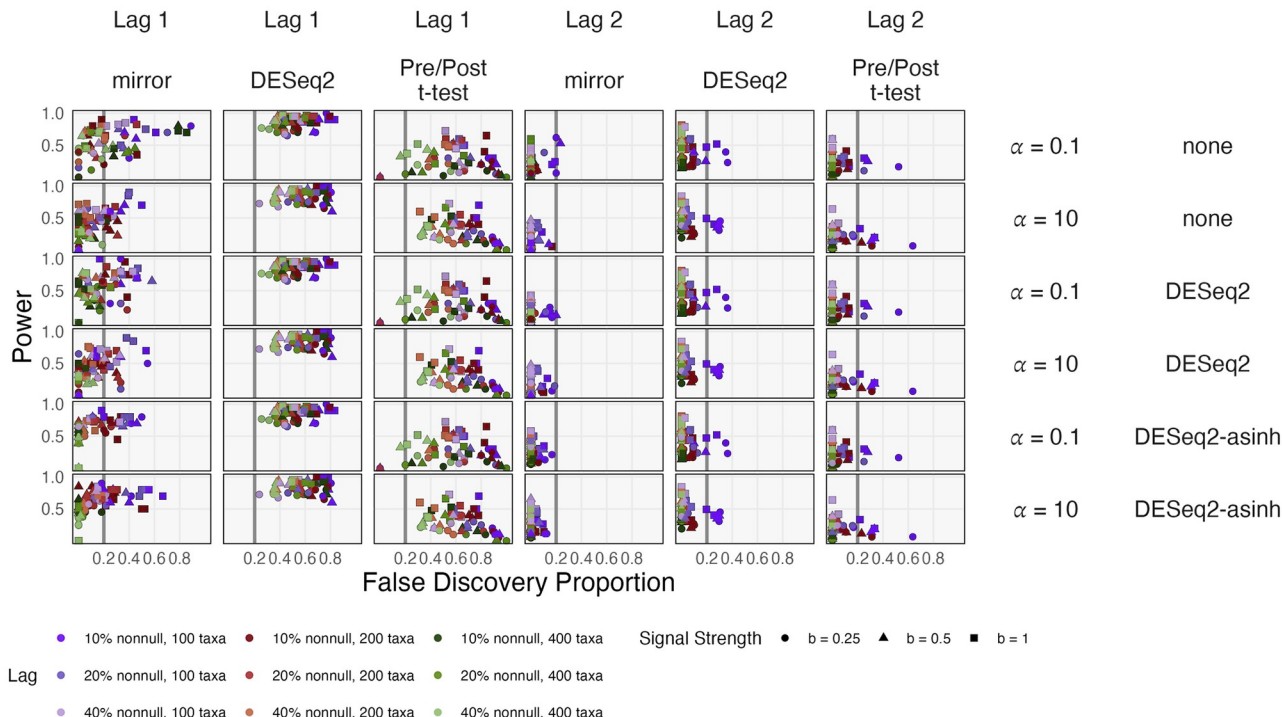

**Fig 6. Inferential performance in the simulation experiment.** Rows encode normalization methods and phylogenetic correlation. Columns have varying lags and compare mirror statistics, DESeq2, and a pre-post *t*-test. Color hue and shade encode the number of taxa and proportion of nonnull hypothesis, respectively. The target FDR has been set to $q = 0.2$ (vertical grey line). DESeq2 lacks FDR control for lag one effects in any simulation context. mbtransfer's mirror algorithm controls the FDR when given DESeq2-asinh transformed data and sufficiently many taxa.

effectively control the FDR when the number of taxa is large and the data have been DESeq2-asinh normalized. This is consistent with the improved forecasting performance for transformed data and with Proposition 3.3 of [12], which guarantees FDR control asymptotically as the number of hypotheses increases. We attribute the reliability and power of mirror statistics to the fact that its false discovery rate control is always adapted to the dataset from which splits are drawn rather than a potentially misspecified probabilistic model.

## Case studies

Next, we investigated how to use mbtransfer with datasets from two human microbiome studies. These datasets include experimentally-defined interventions that arise naturally in prospective studies. Despite differences in taxonomic richness, the number of subjects, and the total number of time points, both explore how environmental change affects the composition of the microbiome.

**Diet and the gut microbiome.** The study [1] considered the human gut microbiome's response to brief dietary interventions. Twenty participants were recruited and randomly assigned to "plant" or "animal" interventions, where they were required to follow a plant- or animal-based diet for five days. Samples were collected for two weeks around the intervention, typically at a daily frequency, yielding 8 and 15 samples per participant due to occasionally missed time points. We use cubic spline interpolation to evenly interpolate time points onto a daily grid, motivated by [36]. Initially, the data contained 17310 taxa. Following the simulation results, we DESeq2-asinh normalized the data. We further filtered to retain taxa present in at

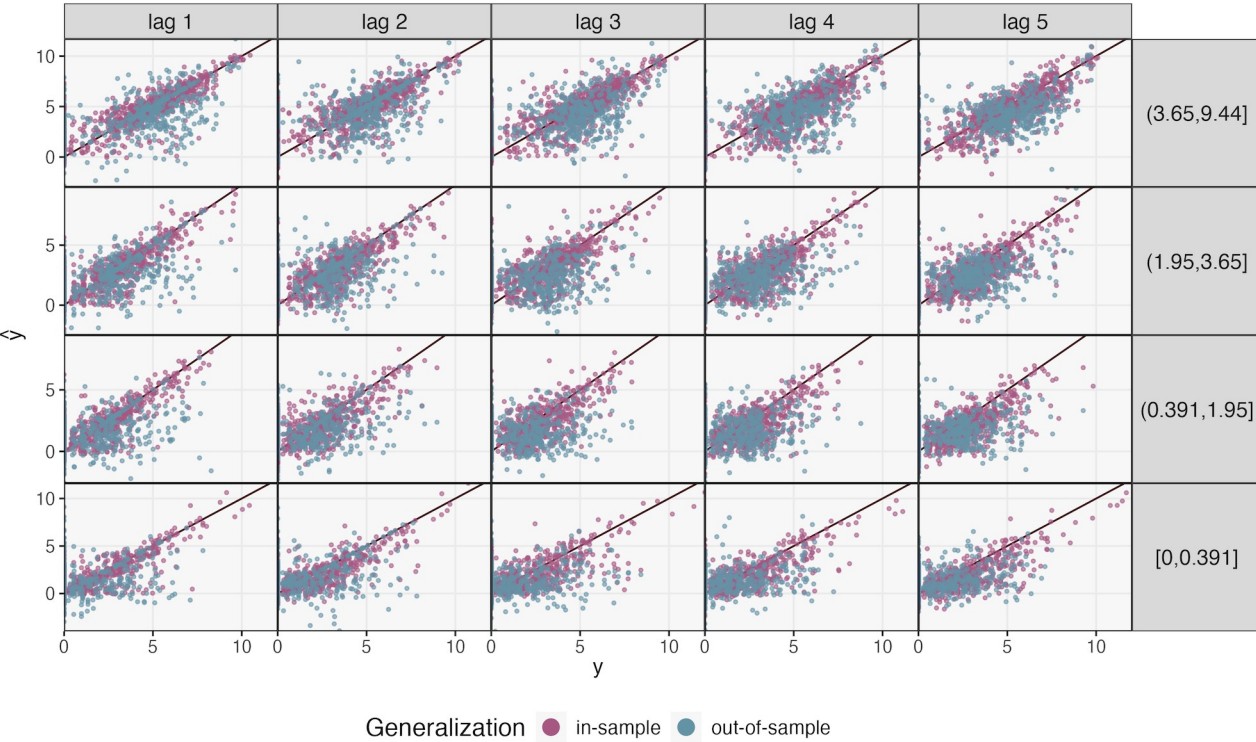

**Fig 7. mbtransfer forecasting error on the diet intervention dataset.** The *y*-axis is faceted by quantiles of abundance and the *x*-axis is faceted by time horizon *h*. In-sample error refers to errors made at new timepoints for individuals who appeared in the training data, while out-of-sample predictions are made on individuals absent from training. Performance is strongest in shorter time horizons and for more abundant taxa.

least 40% of the samples, resulting in 191 taxa. This reduction allowed us to focus our analysis on the "core" microbiome [37, 38].

We used mbtransfer with $P = Q = 2$ and $\mathbf{w}_t^{(i)} = (\mathbb{I}(t \in \text{Animal Diet Shift}), \mathbb{I}(t \in \text{Plant Diet Shift})) \in [0, 1]^2$ to fit our model, creating two intervention series. These series may lie between 0 and 1 due to interpolated time points lying in transitions between active and inactive periods. No additional subject-level covariates $z^{(i)}$ were available. Fig 7 shows in- and out-of-sample forecasts. Forecasting performance deteriorated out-of-sample, highlighting the between-participant heterogeneity in microbiome profiles, particularly within the lowest quantile of abundance. In-sample correlations all lied between $\hat{\rho} \in [0.731, 0.896]$. In contrast, out-of-sample correlation ranged from $\hat{\rho} = 0.123$ (low abundance, lag $h = 3$) to 0.558 (high abundance, $h = 1$), see Table A in S2 Text for details. Forecasting performance was highest on shorter time horizons and for the most abundant taxa. It is possible that, in rare taxa, a prediction model that accounts for sparsity may exhibit higher accuracy. For reference, we generated analogous forecasts with MDSINE2 and fido (Tables B—C and Figs H—I) in S2 Text. Within previously observed subjects, mbtransfer is the top performer, except in three settings with long time horizons and lower quantiles of abundance, where fido performs strongest. In contrast, on previously unobserved subjects, fido outperforms other methods, apart from two settings where MDSINE2 is preferred. This performance aligns with the simulation, which found that mbtransfer can outperform fido when the signal-to-noise ratio is high, while fido's evidence sharing enables better performance in lower signal regimes.

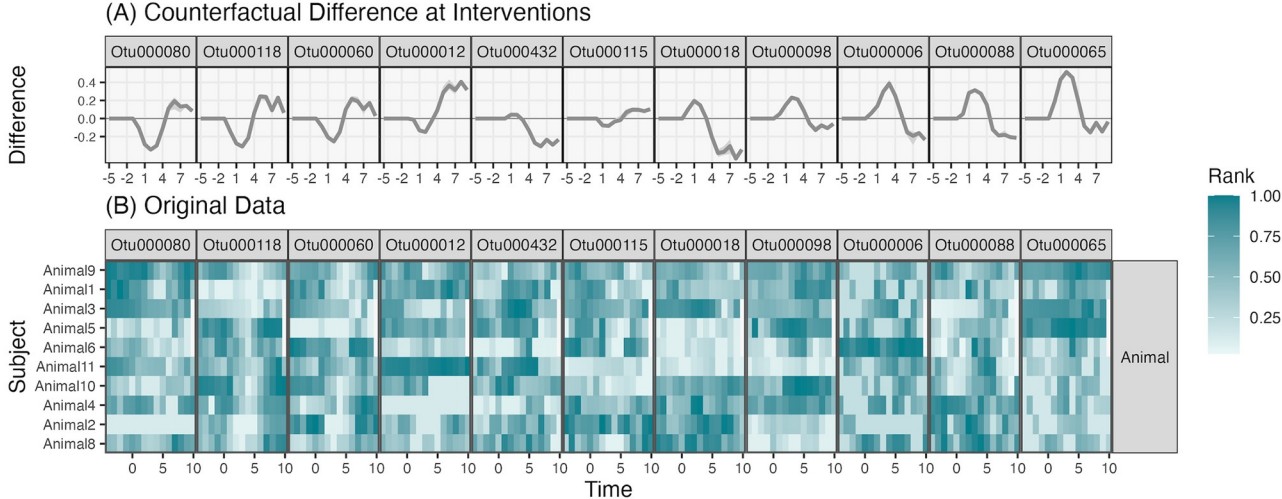

**Fig 8. Intervention-sensitive taxa in the diet study.** (A) Counterfactual difference in simulated trajectories for a subset of the selected taxa in the diet study. (B) Subject-level data from the same taxa, with each row representing a subject and each column a timepoint, potentially interpolated from the original, unevenly sampled measurements. These data are consistent with the interpretations from the counterfactual simulation. For example, OTU000006 often shows transient increases (e.g., Animal1, and Animal6) while OTU000065 has more prolonged departures (e.g., Animal3 and Animal 9).

After assessing model performance, we computed mirror statistics for time lags $h = 1, \ldots, 4$ to evaluate the effect of a four-day shift to an animal diet. Fig L in S2 Text shows the mirror statistic distributions for each taxon and each lag. Increasing magnitudes across lags in certain taxa suggest that the diet intervention effects are not instantaneous but accumulate over consecutive days. Next, Fig 8A shows the median difference between counterfactual trajectories for a subset of significant taxa. Since we cannot visualize all significant taxa simultaneously, we chose this representative subset by applying principal component analysis to the simulated trajectory differences, projecting onto the first component, and selecting every sixth taxon according to that ordering. Some taxa (e.g., OTU000006) exhibited immediate but transient effects, while others (e.g., OTU000065) showed gradual but sustained changes. Further, in several taxa (e.g., OTU000118, OTU000012), an initial decrease was followed by a long-run increase, which is supported by the associated participant-level data. Across taxa in Fig 8A, the ribbons for the birth control and no birth control groups generally overlap. Since these ribbons reflect the range of the estimated counterfactual difference across those groups, this implies that the model did not detect a relationship between the intervention's effect on microbial composition and birth control status. Therefore, the effect of the birth event may be uniform across baseline community profiles. The main benefit of a transfer function modeling approach is the model's capacity to learn different shapes of counterfactual trajectories while still maintaining FDR control.

For comparison, the original, interpolated data for a subset of taxa is shown in Fig 8B. These views are consistent with the counterfactual trajectories, but they are less compact and are obfuscated by the higher degree of sampling noise. Our findings align with those of [1], but we can better describe ecosystem dynamics by modeling the temporal relationship between diet intervention and microbiome community profiles.

**Birth and the vaginal microbiome.** We next re-analyzed data from [2], which followed 49 subjects to study how birth influences the composition of the mother's vaginal microbiome. We considered birth as an intervention. We DESeq2-asinh normalized the abundance and

lightly filtered the data to 29 taxa—this small number is a consequence of the low diversity of the vaginal microbiome. There was regular biweekly sampling except near birth, where samples were gathered daily. Thus, we interpolated to a biweekly resolution.

We used mbtransfer with taxonomy and intervention lags $P = Q = 2$ and contraception usage as a subject-level covariate. Fig J in S2 Text shows in- and out-of-sample forecasting accuracy. Compared to Fig 7 in the previous analysis, in and out-of-sample performances are more comparable, reflecting the larger sample size of this study. We also evaluated forecasts using fido; predictions made using MDSINE2 diverge. We set the fido hyperparameters to $\sigma = 10$ and $\rho = 0.1$ to account for the time scale of this problem. Note that some of fido's forecasts for new subjects yielded missing values, and these were imputed with the mean abundance for that taxon among non-missing entries for that subject. Results appear in Tables D—E and Figs J—K in S2 Text. Fido outperforms mbtransfer for longer time horizons (20–40 days) on previously observed subjects. For other settings, mbtransfer yields better forecasts. We speculate that this improvement for out-of-sample forecasts, relative to the diet intervention, is a consequence of the larger sample size for this study.

Fig M in S2 Text shows the taxon-level mirror statistics. Compared to the diet intervention, the birth intervention caused more pervasive shifts. In light of the simulation results, these findings should be considered tentative, because this problem has relatively few taxa. We generated four counterfactual trajectories for all subjects to understand how birth influences individual taxa and whether any effects are modulated by contraception use. Specifically, we computed $\hat{f}(\mathbf{Y}_{t-1}, \tilde{\mathbf{W}}_t, \tilde{z})$ for $\tilde{\mathbf{W}} \in \{\mathbf{1}_Q, \mathbf{0}_Q\}$ representing presence or absence of the birth event and $\tilde{z} \in \{0, 1\}$ denoting re-initiation of contraceptive use following birth. Fig 9A suggests the absence of an interaction with contraception use. This may be a consequence of the fact that 57% of subjects were missing any data on contraception use—though [2] discussed plausible mechanisms for how contraception use can influence the postpartum microbiome, our model did not detect a strong enough association to confirm this hypothesized interaction effect.

Like in the diet intervention, we can distinguish between response trajectories. Members of the genus *Lactobacillus* were clearly depleted, while other taxa appeared to take advantage of

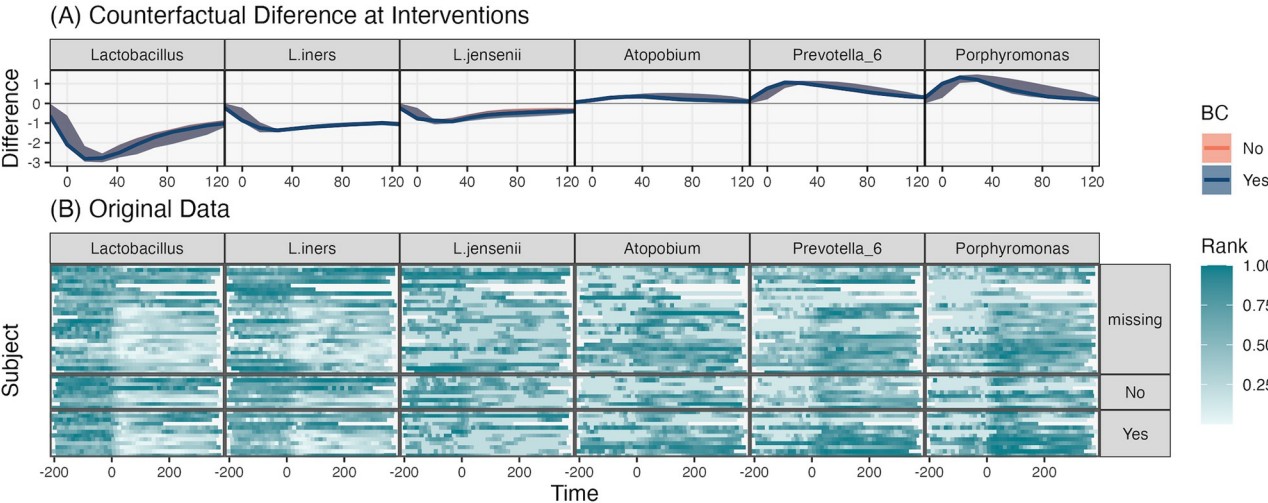

**Fig 9. Intervention-sensitive taxa in the pregnancy study.** (A) Counterfactual differences for a subset of selected taxa from the re-analysis of [2]. Counterfactual differences are computed for each subject in the data, and bands represent the first and third quartiles of differences across subjects. Since the bands for birth control reinitiation overlap, we conclude that the model does not learn the interaction effects between the intervention and contraception use. (B) The corresponding subject-level data grouped by birth control reinitiation survey response. Note that these data have interpolated to the biweekly level to account for uneven sampling times.

novel niches opened up by the postpartum environment. For example, *Porphyromonas* appeared briefly during the same window that the *Lactobacilli* disappeared. Fig 9B compares these trajectories with interpolated observed data. As before, we see that the learned trajectories denoised the original data, and consistent with the lack of interaction, we did not observe systematic associations between postpartum community trajectory and contraception use.

## Discussion

mbtransfer adapts transfer function models to the dynamic microbiome context. The approach is flexible and interpretable, enabling intervention analysis without assuming a restrictive functional form and supporting the simulation of counterfactual trajectories. We have complemented this modeling approach with a formal inferential mechanism, leveraging recent advances in selective inference. A simulation study illuminated our method's properties across data-generating settings, and our data analysis highlighted its practical application in contrasting microbiome studies.

We envision several avenues for future research. First, while our focus has been on developing mirror statistics to detect intervention effects, the same approach could be extended to support the inference of microbe-microbe and host-microbe relationships. The partial dependence profiles in Fig C in S2 Text suggests that mirror statistics could be a promising approach for FDR-guaranteed inference of microbe interaction networks, and future systematic experiments should explore their validity and power. Our simulation study revealed that forecasting performance depends on normalization strategy, and identifying the optimal normalization for transfer function modeling or understanding whether it is possible to bypass it entirely is an open problem [39, 40]. The construction of mirror statistics via partial dependence profiles depends only on having access to a simulator $\mathbf{f}$ that can generate hypothetical responses, and it could be used to contrast alternative initial states $\mathbf{y}_t$ or host features $\mathbf{z}$. Second, developing a transfer function model that learns the entire distribution of responses $p(\mathbf{y}_t|\mathbf{Y}_{t-1}, \mathbf{W}_t, \mathbf{z})$ would be valuable. This probabilistic analog could enable design of interventions where moderate and consistent effects are preferable to strong but erratic ones [41–43]. Finally, extending our approach to continuous-time autoregressive processes would allow irregular sampling frequencies, eliminating the need for interpolation.

We have introduced a flexible but rigorous approach to a recurring microbiome data analysis challenge: How can we quantify the influence of environmental shifts on a microbial ecosystem? Our transfer function perspective has guided the proposed intervention analysis, and we linked the resulting nonlinear models with modern computational inference based on mirror statistics. This facilitates the stability and attribution analysis critical for moving beyond purely descriptive conclusions [44, 45]. As microbiome studies continue to investigate more nuanced questions about ecosystem dynamics, similarly formal simulation and inference methods will play an essential role.

## Supporting information

**S1 Text. Additional discussion.** Discussion of simulation setup, including details of the data generation mechanism and hyperparameter grid used to create contrasting datasets. It also provides mathematical details of the evaluation criteria and an overview of the MDSINE2 and fido algorithms.
(PDF)

**S2 Text. Additional tables and figures.** Numerical summaries and visualization of other settings in the simulation and case studies.
(PDF)

## Acknowledgments

We appreciate the computational support by staff at the Center for High Throughput Computing [46] at the University of Wisconsin-Madison, on which all simulation experiments were run.

## Author Contributions

**Conceptualization:** Kris Sankaran.

**Data curation:** Kris Sankaran, Pratheepa Jeganathan.

**Formal analysis:** Kris Sankaran, Pratheepa Jeganathan.

**Funding acquisition:** Kris Sankaran, Pratheepa Jeganathan.

**Investigation:** Kris Sankaran, Pratheepa Jeganathan.

**Methodology:** Kris Sankaran, Pratheepa Jeganathan.

**Software:** Kris Sankaran.

**Validation:** Kris Sankaran.

**Visualization:** Kris Sankaran, Pratheepa Jeganathan.

**Writing – original draft:** Kris Sankaran, Pratheepa Jeganathan.

**Writing – review & editing:** Kris Sankaran, Pratheepa Jeganathan.

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
