## [Decision Letter · Decision Letter 0]

13 Mar 2024

Dear Dr Jeganathan,

Thank you very much for submitting your manuscript "Microbiome intervention analysis using transfer functions and mirror statistics" for consideration at PLOS Computational Biology.

As with all papers reviewed by the journal, your manuscript was reviewed by members of the editorial board and by several independent reviewers. In light of the reviews (below this email), we would like to invite the resubmission of a significantly-revised version that takes into account the reviewers' comments.

We cannot make any decision about publication until we have seen the revised manuscript and your response to the reviewers' comments. Your revised manuscript is also likely to be sent to reviewers for further evaluation.

Sincerely,

Niranjan Nagarajan

Academic Editor

PLOS Computational Biology

Pedro Mendes

Section Editor

PLOS Computational Biology

Reviewer's Responses to Questions

**Comments to the Authors:**

Reviewer #1: In this article, Sankaran and Jeganathan propose mirror statistics to control FDR when estimating the effect of intervention on microbial abundance using gradient boosting models. This is an interesting and novel statistical solution to a common scientific question. This will be a nice contribution to the literature if the authors can provide additional results and discussion addressing potential limitations of their approach. My major concerns relate to violations of the assumptions required for FDR control with mirror statistics. I have more minor comments especially related to how they discuss fido and mdsine2 as comparisons.

FDR control via mirror statistics is only guaranteed when key assumptions are satisfied. As the authors state, one assumption is the fact that the distribution of mirror statistics is symmetric about zero under the null. Another assumption, which the authors do not state, is that the mirror statistics are weakly correlated (Assumption 2.2 in Dai et al.). I am concerned that, in many practical settings, both assumptions may be violated in the authors applications. As I expand upon below, the authors should provide a more thorough study and discussion of when their method works and when it does not. So long as the authors provide an accurate picture of the limitations of their method, regardless of the results, I think this manuscript will be useful to the field. That said, in its current form these potential limitations are not adequately explored.

Starting with the correlation assumption, there are two well known sources of strong correlations in microbiome data. One is the phylogenetic relatedness of taxa which will almost certainly lead to non-trivial correlations in the proposed mirror statistic. Second is the spurious correlations that can be induced into the data through the arbitrary sequencing depth of the measurement process. Regarding this second source, while the authors state that the data y is normalized, it is non-trivial (and I suspect not possible) to ensure that a chosen normalization will decorrelate the mirror statistics. To address this comment, I encourage the authors to expand their discussion of these assumptions to the manuscript and expand their simulation studies in two ways to provide at least some minimal degree of empirical results as to the robustness of their proposed approach to violations in these assumptions. First, the authors could incorporate varying degrees of phylogenetic structure between the taxa (e.g., phylogenetically patterned correlations in errors or in other confounding effects that might be present under the null). I would encourage the authors to explore how FDR might change as a function of that phylogenetic correlation (e.g., we expect more phylogenetic correlations when analyses are preformed at the sequence variant level than at higher taxonomic/phylogenetic levels). Second, the authors could, and likely should, expand their simulation to incorporate the arbitrary sequencing depth artifacts present in microbiome data. The counts y_i are not independent as the simulation suggests. I suspect that if the negative binomial was replaced with a multinomial (or the counts y from the negative binomial were resampled multivariate hypergeometric to an arbitrary depth) then the authors model will see elevated rates of false positives. These problems will likely become most noticeable when then intervention causes the total microbial load to shift (see McGovern et al., 2023).

Regarding the symmetry assumption, it is less clear how the authors could explore model violations. At present the authors simply state they assume the mirror statistics are symmetric about zero under the null. Notably this seems to rest on an assumption that the error \\(\\epsilon\\) are symmetric about zero. As it is currently written, I do not understand why such an assumption would make sense. For context, Dai et al., make a strong argument that, under a wide range of settings, this assumption is likely true for certain mirror statistics based on linear model or graphical model estimates. The gradient boosting methods proposed are non-trivial and I do not think that prior arguments by Dai can be applied here. This needs further discussion and study. Doesn't a number of panels in figure 6 suggest that this assumption may be violated in practice?

Clarifications are needed to accurately portray the current status quo. To clearly state up front, the authors do not need to change their simulations but simply to add discussions about the limits of how they are using the fido and MDSINE2 tools as comparisons. It is reasonable to use fido and MDSINE2 as a comparison here as there are few potential alternatives and both are used for similar purposes in the literature. Starting with fido. Fido is not designed to perform this task and, as the main author of that package (Justin Silverman), I would not recommend it be used in that way. In fact, in our original paper we explicitly state that this is a method for log-ratio estimation. In other words, fido is designed to estimate linear and non-linear trends in the relative abundances (or log-ratio abundances) of taxa and not their absolute abundance. It is only recently (Nixon et al. 2023, and McGovern et al. 2023) that we have started bridging the gap between estimation of log-ratios and absolute abundances. In short, when you give fido data with absolute abundance information (e.g., negative binomial data rather than multinomial data) fido basically just throws it away as its expecting the data has arbitrary scale. As a result, the authors should report how they are comparing estimates of log-ratios to their ground truth which is basically changes in absolute abundances. Are they using CLR coordinates for this comparison? Others do this but we (and others) have repeatedly expressed concern with this practice (Nixon et al. 2023, and McGovern et al. 2023, or just about any article by Pawlowsky-Glahn or Egozcue). Other non-trivial features that should be reported include the choice of kernel in fido. I am going to guess the authors are just using an RBF kernel in the basset model in fido. This is a common choice but I expect that other kernels might have better performance (again fine to leave as is and just explain choices). Both MDSINE2 and fido are Bayesian models. The choice of prior is non-trivial and should be reported. Note the default priors in fido come with an explicit warning to users urging them to think more deeply about their problem and design appropriate priors that reflect biological knowledge. For MDSINE2, the model was designed to also use qPCR measurements to bridge the gap between relative and absolute estimation. How are the authors then using MDSINE2 here without simulating qPCR data? Finally, as a smaller point, it seems wrong to say that a limitation of fido is its computational speed (page 3) when later (page 9) you say that fidowas comparable and even in once instance faster than mbtransfer.

I think introducing mirror statistics to the field is important as I think there are multiple potential uses for them in the field. That said, the authors barely introduce them or explain their novelty. I think adding a paragraph somewhere providing a bit more background knowledge to the reader would likely improve the impact of the authors work.

On page 3 the PDF seems to be malformed and the sentence starting with "Nevertheless, these methods" is oddly bolded on not on any particular line. Same with the word "profiles a few lines down".

On line 67 you say the errors need to be symmetric but later you say they need to be symmetric and centered about zero. I think the "about zero" likely needs to be added on that line. As this is important and implies absence of systematic bias in the estimator.

"timepoints that are at least max(P,Q) apart" I didn't understand what the authors are trying to say with this phrase (page 5).

Reviewer #2: Manuscript by Sankaran & Jeganathan introduces a new model, mbtransfer, based on transfer function for modeling microbial community dynamics in response to external perturbation. The model allow user to fit and “forecast” the temporal dynamics of the microbiome. The method also implements the mirror statistics to select significantly perturbed taxa. Overall this is a well written paper and would be useful for modelling microbial dynamics in a continuous fashion, especially when the response to external perturbation has a delayed effect. An R package and well documented tutorials on the case studies were provided. I was able to follow the tutorial easily.

I have some minor comments that I would like the authors to clarify:

1. Page 4: “we extract non-overlapping taxonomic and intervention histories”. What does “non-overlapping” mean here?

2. Page 11: “the out-of-sample forecasts showed a clear correlation with the truth”. It would be more rigorous to show the statistics here.

3. Page 11: “Across taxa, we found that the first and third quartiles of the counterfactual differences tended to agree”. It is unclear to me what the authors mean here and why it suggests that “the model did not detect interactions between the intervention effects and microbial composition”.

4. For mirror statistics, is that possible to compare it with a simple baseline statistics testing the abundance before and after perturbation?

Reviewer #3: Uploaded as attachment

**Have the authors made all data and (if applicable) computational code underlying the findings in their manuscript fully available?**

Reviewer #1: Yes

Reviewer #2: Yes

Reviewer #3: Yes

PLOS authors have the option to publish the peer review history of their article (what does this mean?). If published, this will include your full peer review and any attached files.

Reviewer #1: **Yes: **Justin D Silverman

Reviewer #2: No

Reviewer #3: No
---

## [Decision Letter · Decision Letter 1]

27 May 2024

Dear Dr Jeganathan,

We are pleased to inform you that your manuscript 'mbtransfer: Microbiome intervention analysis using transfer functions and mirror statistics' has been provisionally accepted for publication in PLOS Computational Biology.

Best regards,

Niranjan Nagarajan

Academic Editor

PLOS Computational Biology

Pedro Mendes

Section Editor

PLOS Computational Biology

Reviewer's Responses to Questions

**Comments to the Authors:**

Reviewer #1: The authors have addressed my comments appropriately. In particular I think the discussion of mirror statistics (the added background) adds substantially. I think the presentation of their method is also much improved in the revision and provides readers with a greater ability to judge the benefits and limitations of the proposed approach.

Reviewer #2: The authors had addressed my comments.

**Have the authors made all data and (if applicable) computational code underlying the findings in their manuscript fully available?**

Reviewer #1: Yes

Reviewer #2: Yes

PLOS authors have the option to publish the peer review history of their article (what does this mean?). If published, this will include your full peer review and any attached files.

Reviewer #1: **Yes: **Justin D Silverman

Reviewer #2: No

---

## [Editor Report · Acceptance letter]

8 Jun 2024

PCOMPBIOL-D-24-00098R1 

mbtransfer: Microbiome intervention analysis using transfer functions and mirror statistics

Dear Dr Jeganathan,

I am pleased to inform you that your manuscript has been formally accepted for publication in PLOS Computational Biology. Your manuscript is now with our production department and you will be notified of the publication date in due course.

With kind regards,

Zsofia Freund
